# Cytotoxicity of Acrylic Resins, Particulate Filler Composite Resin and Thermoplastic Material in Artificial Saliva with and without Melatonin

**DOI:** 10.3390/ma15041457

**Published:** 2022-02-16

**Authors:** Seda Cengiz, Neslin Velioğlu, Murat İnanç Cengiz, Fehiye Çakmak Özlü, Ahmet Ugur Akbal, Ahmet Yılmaz Çoban, Mutlu Özcan

**Affiliations:** 1Department of Prosthodontics, Faculty of Dentistry, Zonguldak Bülent Ecevit University, Zonguldak 67600, Turkey; 2Department of Prosthodontics, Navadent Oral and Dental Health Policlinic, Zonguldak 67000, Turkey; nes_ay_vel@yahoo.com.tr; 3Department of Periodontology, Faculty of Dentistry, Zonguldak Bülent Ecevit University, Zonguldak 67600, Turkey; minanc.cengiz@beun.edu.tr; 4Department of Orthodontics, Faculty of Dentistry, Ondokuz Mayıs University, Samsun 55270, Turkey; ckfethiye@hotmail.com; 5Unit of Infectious Diseases, Samsun Health Directorate, Samsun 55060, Turkey; auakbal@gmail.com; 6Department of Medical Biotechnology, Institute of Health Sciences, Akdeniz University, Antalya 07058, Turkey; cobanay2003@gmail.com; 7Department of Nutrition and Dietetics, Faculty of Health Sciences, Akdeniz University, Antalya 07058, Turkey; 8Tuberculosis Research Center, Akdeniz University, Antalya 07058, Turkey; 9Center for Dental and Oral Medicine, Division of Dental Biomaterials, Clinic for Reconstructive Dentistry, University of Zurich, 8032 Zurich, Switzerland; mutluozcan@hotmail.com

**Keywords:** melatonin, cytotoxicity, artificial saliva, dental materials

## Abstract

There is limited information on the effect of melatonin on the cytotoxicity of dental materials. The study evaluated the cytotoxic effects of heat- and auto-polymerized acrylic resin, particulate filler composite resin and a thermoplastic material on L-929 fibroblast cell viability at different incubation periods in artificial saliva without and with melatonin. Disk-shaped specimens were prepared according to each manufacturer’s instructions and divided into two groups to be stored either in artificial saliva (AS) and AS with melatonin (ASM). The measurements were performed using an MTT (3-(4,5)-dimethylthiazol-2-yl)-2,5-diphenyl tetrazoliumbromide) assay, in which the L-929 mouse fibroblasts cell culture was used. For the MTT test, extracts were examined at 1, 24, 72 h and 1 and 2 weeks. Data were analyzed using 3-way ANOVA and Tukey’s tests. No significant difference was found between groups AS and ASM (F = 0.796; *p* = 0.373). Incubation period significantly affected all materials tested (*p* < 0.001). Storing resin-based materials in artificial saliva with melatonin solution for 24 h may reduce cytotoxic effects on the fibroblast cells for which the highest effect was observed. Soaking resin prosthesis or orthodontic appliances in artificial saliva with melatonin at least 24 h before intraoral use or rinsing medium containing melatonin may be recommended for decreasing the cytotoxicity of dental resin materials.

## 1. Introduction

Melatonin (5-methoxy-N-acetyltryptamine) defined as a neurotransmitter-like compound is mainly produced in the pineal gland and also other tissues in the location of extrapineal areas such as the retina, harderian glands, hypotalamus and gut [1]. Numerous studies reported on melatonin and its effect on damaged DNA samples induced by chemicals and radiation [2,3,4,5]. In addition to DNA protection, melatonin also displays strong antioxidative characteristics and acts as a potential antiapoptotic agent [6,7,8].

Dental treatment with the use of composites, various of restorative materials, orthodontic braces, titanium implants, or whitening agents can disturb the oral redox homeostasis by affecting the antioxidant barrier and increasing oxidative damage to salivary proteins, lipids, and DNA. Oxidative stress is a redox homeostasis disorder that results in oxidation of cell components and cell metabolism disorder. It can be defined as a disturbance in the balance of oxygen–nitrogen free radicals and neutralization factors. Enzymatic and non-enzymatic antioxidant systems protect cells against harmful effects of oral redox homeostasis. The glutathione system, albumins, lactoferrin, ascorbic acid, uric acid and melatonin are the most important non-enzymatic antioxidant systems. It was also reported that melatonin showed a positive effect on monocytes, cytokines and fibroblasts having an impact on angiogenesis [9,10,11,12,13,14].

Methacrylate polymers are the main structure of contemporary dental restorative materials. However, due to an incomplete polymerization process while mechanical shearing, methacrylate monomers are released into systemic circulation during polymerization through the oral cavity or the pulp. According to in vivo and in vitro cytotoxicity and genotoxicity studies, such monomers present toxic properties and a considerable portion of these effects is underlined by the oxidative reaction of these compounds [15,16].

Polymeric forms of methacrylates are generally used in prosthodontics and orthodontics [17]. The in situ polymerization process is never complete and incomplete polymerization may result in dispersion of methacryclate monomers to the circulation system through the oral cavity. Since monomers may cause adverse biological effects, including cytotoxicity and genotoxicity, anti-inflammatory, anti-oncologic antioxidative effects melatonin may serve as a preventive agent against methacrylate compounds [17,18,19,20,21].

In most of the studies the MTT (tetrazolium salt 3-[4,5-dimethylthiazol-2-yl]-2,5- diphenyltetrazolium bromide) test was used in order to evaluate cytotoxicity of dental materials. The MTT assay delivers information on cell viability by evaluating the alterations in mitochondrial succinate dehydrogenase activities [22,23]. In this test, methylthiazol tetrazolium is reduced and a purple-colored formazan was formed. It has been previously shown that active cells produce more formazan than resting cells and therefore by measuring formazan-level changes, it is also possible to measure the cell metabolism [22].

The objective of this study, therefore, was to investigate the cytotoxic effects of heat- and auto-polymerized acrylic resin, resin composite and a thermoplastic material on L-929 fibroblast cell viability at different incubation periods in AS and ASM. The null hypothesis was that there is no difference in cytotoxicity between AS and ASM. The second null hypothesis was that there is no difference in cytotoxicity between the different materials in different incubation periods.

## 2. Materials and Methods

Table 1 shows the brand names, abbreviations, chemical compositions and manufacturers of the materials used in this study. A stainless-steel mold (diameter: 10 mm; thickness:1 mm) was used to fabricate the specimens. Ten disk-shaped specimens were prepared from each of the nine materials. The stainless steel mold was pressed by two glass plates both bottom and top and for extruding excess resin, finger pressure was applied.

### 2.1. Specimen Preparation

Acrylic resin: the acrylic resin was placed between two glass plates. Specimens were manufactured according to each manufacturer’s instructions. V, O, I specimens were polymerized in a polymerization oven (Polyclav, Dentaurum, Ispringen, Germany). P, M specimens were polymerized in the muffle furnace (Ermetal, Balıkesir, Turkey).

Particulate filler resin composite: particulate filler resin composites (S, A and T) were placed between two glass plates. The glass plate was removed prior to polymerization in their corresponding photo-polymerization unit. Specimens of material S were polymerized in a photo-polymerization unit (Heraflash, Hanau, Germany). Specimens of material A were polymerized in heat and a photo-polymerization unit (Targis Power Upgrade, Schaan, Liechtenstein), and specimens of material T were polymerized in heat and a photo-polymerization unit under pressure (Tescera ATL, Schaumburg, IL, USA).

Thermoplastic material: a stainless steel mold was placed above the metal layer and the thermoplastic material was layered and polymerized in its corresponding polymerization device (Biostar, Scheu, Iserlohn, Germany). Then, specimen surfaces were ground finished with a series of silicone carbide papers (280-, 400-, 600-, 800- and 1000- grit) (3M ESPE, St. Paul, MN, USA) for 10 s in a polishing machine (Buehler Metaserv, Buehler, Germany) at 300 rpm providing a flat and uniform surface. They were ultrasonically cleaned for 3 min in deionized water and dried with air. Ethylene oxide gas was used to sterilize the specimens.

### 2.2. Storage and Test Medium

The specimens were divided into two groups to be stored either in artificial saliva (AS) and AS with melatonin (ASM). Each specimen in AS group were stored in 1 mL AS containing 4.1 mM KH_2_PO_4_, 4.0 mM Na_2_HPO_4_, 24.8 mM KHCO_3_, 16.5 mM NaCl, 0.25 mM MgCl_2_, 4.1 mM citric acid, and 2.5 mM CaCl_2._ The pH of the AS was adjusted to 6.7 which was sterilized by filtration before use.

Each specimen in ASM was stored in melatonin solution consisting 50 mM which was dissolved in aqueous solution from a 10 mM stock. Melatonin suspension with a final concentration of 0.025 M was obtained by adding 100 μL of melatonin (Sigma-Aldrich Corp., St. Louis, MO, USA) dissolved in ethanol to 50 mL of the AS. Dulbecco’s Modified Eagle Medium (DMEM; 1:1; Sigma, St. Louis, MO, USA) supplemented with 10% fetal bovine serum (FBS; Biochrom, Berlin, Germany) was used as test medium.

### 2.3. Cell Culture

Mouse fibroblast cell culture (L-929 An 2 HÜKÜK 95030802: Foot and Mouth Disease Institute, Ankara, Turkey) was used in this study. Trypsination was performed by 0.25% concentration of 1 mL of trypsin (Sigma-Aldrich Corp. St. Louis, MO, USA) in a 75 cm flask. After washing trypsin, 2 mL of trypsin was added and the cells were detached after 5–10 min of incubation. Following the detachment of the cells, 4 mL of DMEM with FBS was added. After centrifugation at 800 rpm for 5 min, the upper medium-trypsin part was removed. The pellet was used to obtain a suspension and cultured into the flasks. Cell cultures of 3 × 10^4^ cell/mL were used to prepare the suspensions. Cells were counted using a hemocytometer (Thoma chamber) and the concentration was calculated by a formula of (number of cells/counted square) × 100,000. Cell culture suspensions were incubated for 24 h after addition to the test wells. Then, the culture medium was removed and test extracts were added to the wells. Test extracts were retrieved after at 1, 24, 72 h and 1 and 2 weeks. Prior to the MTT test (concentration: 5 mg/mL), the L-929 cell culture suspension was cultured with test extracts during these time points.

Absorption index was calculated as follows:

Absorption index of AS: Example (AS)/Control × 100 

Absorption index of ASM: Example (ASM)/Control × 100 

### 2.4. Cytotoxicity Test

Cytotoxicity tests were performed in two stages. Firstly, the disks of each material were divided into two groups and were placed in the wells of the plates. AS was added to the wells of half of the group for each material type, and ASM was added to the wells of the second half of specimens. The plates were incubated for 1 h at 37 °C and the solutions in the wells were retrieved after incubation. DMEM of 1 mL of FBS was added to each well. The plates were kept at 37 °C in the incubator and 500 µL of test extracts were taken at 1, 24, 72 h and 1 and 2 weeks and stored at −20 °C. 

At the second stage, 96-well plates were used for the extracts with melatonin (ASM) and those without melatonin (AS). Three wells in the plates were used for each of the test materials while nine wells were conserved for the control group. Each plate was filled merely with the medium. Cell culture of 100 µL suspension was distributed into each well and the medium was cautiously removed from the wells after 24 h of incubation. After the incubation process, 100 µL test extract was inserted into each well. Test extracts were retrieved at 1, 24, 72 h and 1 and 2 weeks for the MMT test. MTT (5 mg/mL concentrated 100 µL) (Sigma Aldrich, Darmstadt, Germany) was added to the wells and removed after 2 h of incubation at 37 °C before 100 µL dimethyl sulfoxide was added. After agitation for 15 min, the absorbance was measured on a 570 nm absorbance-plate reader (Thermo Scientific Multiskan EX, Waltham, MA, USA). Metabolically active viable cells convert MTT into a purple-colored formazan with a maximum absorbance near 570 nm.

### 2.5. Statistical Analysis 

Statistical analysis was performed using a software package (Statistical Package for Social Sciences, SPSS 15 software, SPSS Inc., Chicago, IL, USA). Considering three factorial variables (Level 1: AS vs. ASM; Level 2: 11 materials; Level 3: 5 timepoints- 1, 24, 72 h, 1, 2 weeks), data from absorbant index results were analyzed using three-way ANOVA and Tukey’s tests at a significance level of *p* < 0.05. 

There were three factors in the experiment. 1: AS and ASM, 2: dental materials (negative control, T, A, B, P, O, M, S, I, V, positive control). 3: time factor. The total number of application combinations was 11 × 2 × 3 = 66. Results were obtained using a five-fold dwell time. Measurements over time were not used on the same experimental unit, but a separate set of subjects was used for each time period. Each combination was repeated three times. Therefore, there are (2 × 11 × 5) × 3 = 330 observations in the experiment. The trial was set up according to the factorial trial order. The measured property is the absorbant value that defines the toxic effect. The aim is to reveal the effect of melatonin on these absorbant values, the effect of the materials used, the time and the interaction effects of melatonin–material, melatonin–time, material–time and melatonin–material–time.

## 3. Results

Cytotoxicity results showed no significant difference between the AS and ASM groups (F = 0.796; *p* = 0.373) directly. However, there was a significant difference for both tested materials at each incubation period (*p* < 0.001). Material type (F = 2.212; *p* < 0.05) and incubation periods (F = 18.565; *p* < 0.001) significantly affected the results for AS and ASM. Interaction terms between the tested materials and the incubation periods were not significant (F = 0.864; *p* = 0.691). Three-factor interaction (group, material and incubation time) was not significant (F = 1.221; *p* = 0.196) (Table 2).

The absorbant index values for the AS and ASM groups are listed in Table 3 and Table 4. Estimated marginal means of absorbant index of the groups in the incubation periods were shown in Figure 1. Melatonin shows its maximum effect in the first 24 h. Then the effect decreases and shows a sinusoidal change. After 24 h, the absorbant index values decreased (Figure 2).

Although there was not a significant difference between the O, V, I materials in ASM at 1 h incubation period, the absorbant index values increased for M, A, T, B, S and P materials (Table 4).

## 4. Discussion

This study was undertaken in order to investigate the cytotoxic effects of heat- and auto-polymerized acrylic resin, particulate filled resin composite and a thermoplastic material on L-929 fibroblast cell viability at different incubation periods in artificial saliva without and with melatonin. The materials chosen were primarily methacrylate-based dental resins indicated for prosthodontic and orthodontic applications in dentistry. Based on the results of this study, since material type and incubation periods showed a significant effect on the results for AS and ASM, the second null hypothesis could be rejected.

AS containing inorganic components is the basic requirement of an artificial oral environment [24]. Although in vitro cytotoxicity of methacrylate-based dental resin studies with AS were present in the literature, no study considered the effect of melatonin on cytotoxicity. Melatonin exhibits biocompatibility with the oral cavity and this advantage was considered as a protection for harmful effects of dental restorations [15]. The amount of formazan product is generally proportional to the number of active viable cells. When cell metabolism slows down, the amount of MTT reduction per cell will decrease. It will lead to a corruption of linearity between absorbance and cell loss. In this in vitro study, the cytotoxicity of acrylic resin, particulate filler composite resin and thermoplastic material were determined by an MTT test [25] and storage conditions were limited to 14 days. 

Abnormal salivation has been reported in dental patients during oxidative stress occurring. Researchers suggested that the addition of antioxidants to dental materials or adding antioxidant treatment to the dental treatment planning can decrease the oxidative stress level of the oral cavity [9].

Primary cells and continuous cell lines are used in the cell culture studies. Primary cell cultures are obtained by culturing cells separated from tissues and organs for more than 24 h. Pulp fibroblasts are examples of the primary cultured cells. However, it is very difficult to isolate and culture primary cultures from humans. Because primary cultures are taken from different individuals, they reflect functional states differently. Persistent cell lines are transformed primary cells that are capable of proliferating indefinitely and have a more stable phenotype. Although continuous cells cannot maintain all their in vivo properties due to the transformation that occurs, they can be easily propagated. Mouse fibroblast cell culture L-929 is used in many studies as a standardized cell culture for the evaluation of viability and determination of viable cell number [26,27,28,29]. 

Unfortunately, for most of the resin-based materials, the polymerization process is not completed for a long period in oral applications [30,31]. For this reason, cytotoxicity of such materials were evaluated at 1, 24, 72 h and 1 and 2 weeks. Monomers may be released from polymerized materials by the mechanical stress associated with chewing and by the enzymes present in the saliva. On the other hand, polymethacrylates may have ester groups in their structure that hydrolize at their surface which may release monomers into the oral cavity. Methacrylate acid-based monomers are esters and, therefore, they can dissolve in the organism [32].

This study showed that melatonin was found to be effective in AS. When considering time interaction, there was significant difference within the groups. Goldberg’s findings suggest that resin containing materials are more cytotoxic at early intervals [33]. The biological viability of resin materials has often been expected in the first 24 h [34,35]. For the tested resin materials, melatonin might eliminate cytotoxicity especially in the first 24 h. Furthermore, in our study we found a significant difference between material types. At 1 h, all auto-polymerized acrylic resin specimens (M) showed no change in cytotoxicity level. However, heat-polymerized ones, particulate filler composite resins and the thermoplastic materials presented decreased cytotoxicity at 1 h. This finding may be attributed to the effect of melatonin on cytotoxicity in early time periods, playing a role on factors like heat, light and pressure of the polymerization units. In this study, melatonin suspension with a final concentration of 0.025 M was used [36]. Future studies should focus on the dose-dependent effect of melatonin.

## 5. Conclusions

Storing resin-based materials in artificial saliva with melatonin solution for 24 h may reduce cytotoxic effects on the fibroblast cells where the highest effect was observed. Within the limitations of this in vitro study, resin prosthesis and orthodontic appliances could be stored in artificial saliva with melatonin or in mouth rinse containing melatonin for at least 24 h prior to intraoral use in order to decrease their cytotoxic effect.

## Figures and Tables

**Figure 1 materials-15-01457-f001:**
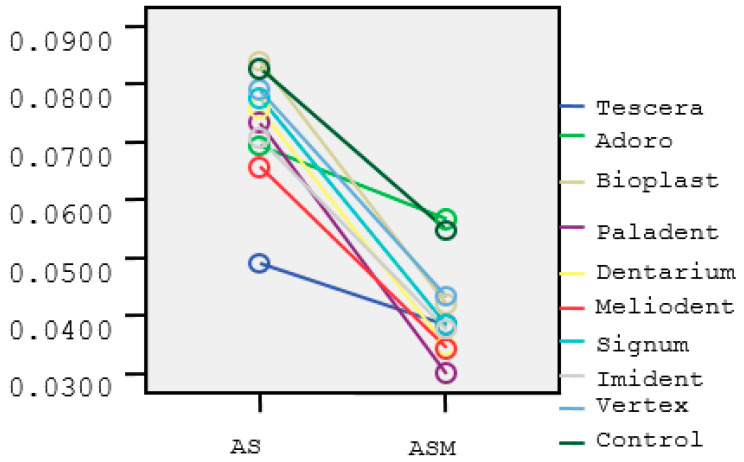
Estimated marginal means of absorbant index of the groups in the incubation periods. Different colors shows the absorbant values of the materials (T, A, B, P, D, M, S, I, V) and the control group.

**Figure 2 materials-15-01457-f002:**
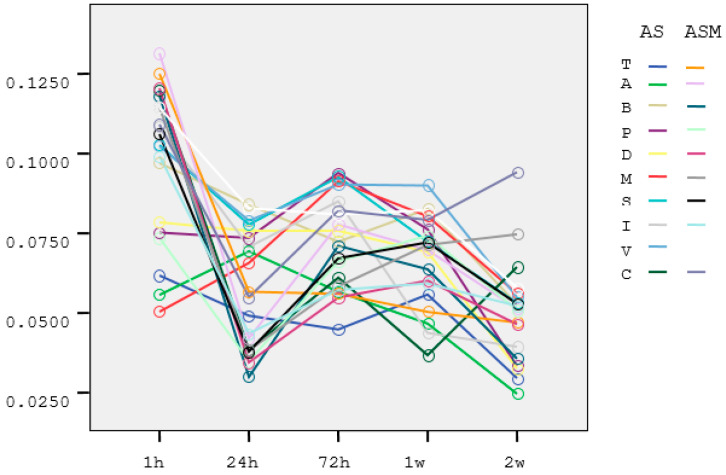
Means of absorbant index at 24 h. Different colors shows the absorbant values of the materials (T, A, B, P, D, M, S, I, V) and the control group in artificial saliva (AS) and artificial saliva with melatonin (ASM).

**Table 1 materials-15-01457-t001:** Brand names, abbreviations, chemical compositions and manufacturers of the materials used in this study.

Materials	Abbreviation	Chemical Composition	Manufacturer
Vertex	V	Powder: Polymethylmethacrylate, liquid; MethylMethacrylate, Crosslinker, Accelerator, UV Absorber	Vertex-Dental, Zeist, The Netherlands
Orthocryl	O	Powder: Polymethylmethacrylate; liquid: Methyl 2-methylprop-2-enote, Methyl 2-methylpropenoate Methylmethylpropenoate Methylmethacrylate	Dentarium, Ispringen, Germany
Imident	I	Power: Polymethylmethacrylate; liquid:MethylMethacrylate, Crosslinker, Accelerator	Imicryl, Konya, Turkey
Paladent	P	Powder: Polymethylmethacrylate; liquid: MethylMethacrylateDimethylacrylate	Med-Dent, Ankara, Turkey
Meliodent	M	Powder: Polymethylmethacrylate; liquid:MethylMethacrylateDimethylacrylate	HeraeusKulzer, Hanau, Germany
Bioplast	B	Ethylenvinylacetat (EVA)	AmBurgberg, Iserlohn, Germany
Signum	S	Bisphenol A-diglycidyl dimethacrylate (Bis-GMA) and Triethylene glycol dimethacrylate (TEGDMA), SiO_2_, Ba-Al-Si	HerausKulzer, Hanau, Germany
Tescera	T	Glass, amorf silica, Etoksilatbisfenol-A-dimethacrylat, bisfenol-A-diglisidilmethacrylate	Bisco, Schaumburg, IL, USA
Adoro	A	UDMA, SiO_2_	Ivoclar-Vivadent, Schaan, Liechtenstein

**Table 2 materials-15-01457-t002:** Three-way analysis of variance (ANOVA) and Tukey’s post-hoc results for comparison of absorbant index values of the tested materials. * shows the interaction between the groups.

Source	Type III Sum of Squares	df	Mean Square	F	*p*
Groups (AS, ASM)	0.000	1	0.000	0.796	0.373
Material	0.019	9	0.002	5.013	0.000
Incubation period	0.082	4	0.020	47.944	0.000
Groups (AS, ASM) * Material	0.008	9	0.001	2.212	0.023
Groups (AS, ASM) * Incubation period	0.032	4	0.008	18.565	0.000
Material * Incubation period	0.013	36	0.000	0.864	0.691
Groups (AS, ASM) * Material * Incubation period	0.019	36	0.001	1.221	0.196
Error	0.085	200			
Total	1.674	300			
Corrected Total	0.259	299			

**Table 3 materials-15-01457-t003:** The mean and standard deviations of absorbant index level per material in artificial saliva (AS). Different letters ^a,b,c^ in the same column indicate significant differences (*p* < 0.05). See Table 1 for group abbreviations.

Materials	1 h	24 h	72 h	1 Week	2 Week	Mean	*p*
**T**	0.06 ± 0.01 ^b^	0.05 ± 0.01 ^c^	0.04 ± 0.01 ^c^	0.06 ± 0.01 ^c^	0.03 ± 0.01 ^c^	0.04 ± 0.01	0.06
**A**	0.06 ± 0.03 ^b^	0.07 ± 0.02 ^c^	0.06 ± 0.01 ^c^	0.05 ± 0.01 ^c^	0.02 ± 0.01 ^c^	0.04 ± 0.01	0.06
**B**	0.1 ± 0.03 ^b^	0.08 ± 0.04 ^c^	0.07 ± 0.03 ^c^	0.08 ± 0.01 ^c^	0.05 ± 0.01 ^c^	0.06 ± 0.01	0.3
**P**	0.08 ± 0.01 ^a^	0.07 ± 0.01 ^a^	0.09 ± 0.01 ^a^	0.08 ± 0.01 ^a^	0.03 ± 0.01 ^b^	0.06 ± 0.01	0.01
**O**	0.08 ± 0.04 ^b^	0.08 ± 0.01 ^c^	0.08 ± 0.01 ^c^	0.07 ± 0.01 ^c^	0.03 ± 0.02 ^c^	0.05 ± 0.01	0.19
**M**	0.06 ± 0.02 ^a^	0.07 ± 0.01 ^a,b^	0.09 ± 0.01 ^b^	0.08 ± 0.01 ^a,b^	0.06 ± 0.01 ^ab^	0.06 ± 0.01	0.02
**S**	0.1 ± 0.02 ^a^	0.08 ± 0.01 ^a,b^	0.09 ± 0.01 ^a,b^	0.07 ± 0.01 ^a,b^	0.05 ± 0.01 ^b^	0.06 ± 0.01	0.02
**I**	0.11 ± 0.02 ^a^	0.08 ± 0.01 ^a,b^	0.09 ± 0.02 ^a,b^	0.09 ± 0.01 ^b^	0.06 ± 0.02 ^b^	0.05 ± 0.02	0.02
**V**	0.10 ± 0.05 ^a^	0.08 ± 0.01 ^a,b^	0.08 ± 0.01 ^a,b^	0.08 ± 0.03 ^b^	0.06 ± 0.02 ^b^	0.06 ± 0.01	0.02

**Table 4 materials-15-01457-t004:** The mean and standard deviations of absorbant index level per material in artificial saliva with melatonin in (ASM). Different letters ^a,b,c,d^ in the same column indicate significant differences (*p* < 0.05). See Table 1 for group abbreviations.

Materials	1 h	24 h	72 h	1 Week	2 Week	Mean	*p*
**T**	0.12 ± 0.02 ^a^	0.05 ± 0.01 ^b^	0.06 ± 0.01 ^b^	0.04 ± 0.01 ^b^	0.04 ± 0.01 ^b^	0.06 ± 0.01	0.009
**A**	0.12 ± 0.02 ^a^	0.04 ± 0.01 ^b^	0.06 ± 0.01 ^b^	0.04 ± 0.01 ^b^	0.06 ± 0.02 ^b^	0.06 ± 0.01	<0.001
**B**	0.13 ± 0.02 ^a^	0.06 ± 0.01 ^b^	0.06 ± 0.01 ^b^	0.05 ± 0.01 ^b^	0.05 ± 0.02 ^b^	0.07 ± 0.01	0.008
**P**	0.13 ± 0.01 ^a^	0.04 ± 0.02 ^b^	0.08 ± 0.02 ^c^	0.07 ± 0.01 ^b,c^	0.05 ± 0.01 ^b,c^	0.07 ± 0.01	0.023
**O**	0.12 ± 0.02 ^b^	0.03 ± 0.01 ^c^	0.07 ± 0.02 ^c^	0.06 ± 0.0 ^d^	0.04 ± 0.01 ^d^	0.06 ± 0.01	0.463
**M**	0.07 ± 0.02 ^a^	0.07 ± 0.01 ^a,b^	0.09 ± 0.01 ^b^	0.08 ± 0.01 ^a,b^	0.06 ± 0.01 ^a,b^	0.06 ± 0.01	0.02
**S**	0.12 ± 0.02 ^a^	0.03 ± 0.01 ^b^	0.06 ± 0.02 ^a,b^	0.06 ± 0.02 ^a,b^	0.05 ± 0.01 ^a,b^	0.06 ± 0.02	0.02
**I**	0.11 ± 0.04 ^b^	0.04 ± 0.01 ^c^	0.06 ± 0.02 ^c^	0.07 ± 0.01 ^d^	0.08 ± 0.03 ^d^	0.07 ± 0.02	0.1
**V**	0.11 ± 0.05 ^c^	0.04 ± 0.01 ^c^	0.07 ± 0.04 ^c^	0.07 ± 0.03 ^d^	0.06 ± 0.02 ^d^	0.07 ± 0.02	0.1

## Data Availability

Not applicable.

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
