# Peer review of "Cytotoxicity of Acrylic Resins, Particulate Filler Composite Resin and Thermoplastic Material in Artificial Saliva with and without Melatonin"

_materials, 2022, doi:10.3390/ma15041457_

Round 1

Reviewer 1 Report

The manuscript entitled "Cytotoxicity of acrylic resins, particulate filler composite resin and thermoplastic material in artificial saliva with and without melatonin" focuses on the cytotoxic effects of heat- and auto-polymerized acrylic resin. The advantage of this work is that a large number of tests are carried out on various types of materials. As many as 9 different materials were examined in this report as shown in Table 1. However, while reading, I highlighted a few points worth clarifying / correcting. They are as follows: 

1) Why were studies on fibroblasts carried out? Why was this type of cell line chosen for research? 

2) The numerical values in table 2 are difficult to understand because, for example, ", 002" how to understand it? in one of the columns the values of p were collected, which are explained above as probability. However, in table 2 p takes the values ".691" when it should not exceed 1. All this should be explained and possibly corrected. 

3) Editorial errors should be corrected throughout the manuscript.

4) In table 3, in some places, the values of standard deviations are similar to the obtained values, which should not appear, because it means that the obtained values are marked with too much error ...  

5) The Conclusions part is far too laconic and must be expanded to include the concrete achievements of this manuscript. 

6) The most recent items are missing in the cited literature, for example, at least from the last 3 years. This should be completed. 

Author Response

Response to Reviewer 1 Comments

Point 1: Why were studies on fibroblasts carried out? Why was this type of cell line chosen for research?

Response 1: The L 929 culture we use is sold commercially and is used for viability assessment and live cell detection.

"Two types of cells are used in the cell culture studies. These are primary cells and continuous cell lines. Primary cell cultures are obtained by culturing cells separated from tissues and organs for more than 24 hours. Pulp fibroblasts are examples of the primary cultured cells. However, it is very difficult to isolate and culture primary cultures from humans. Since primary cultures are taken from different individuals, they reflect functional states differently.

Persistent cell lines are transformed primary cells which are capable of proliferating indefinitely and have a more stable phenotype. Continuous cells cannot maintain all their in vivo properties due to the transformation that occurs. Continuous cell lines can be easily propagated. L 929 is also one of the continuous cell lines frequently used in the research studies. Mouse fibroblast cell culture L-929 is used in many studies as a standardized cell culture for the evaluation of viability and determination of viable cell number. It is also possible to repeat cytotoxicity tests because it is standardized. In addition, fibroblast cell cultures are recommended in the standards such as ISO 7405."

Dental Materials 18 (2002) 318-323 Responses of L929 moue fibroblasts, primary and immortalized bovine dental papillaa,derived cell lines to dental resin components. B. Thonemnn, G. Schmalz, K.A. Hiller, H. Schweikl.

ISO7405

Med Oral Patol Oral Cir Bucal 2007;12:E258-66

Schmalz G, Dorthe Arenholt Bindslev. Biocompatibility of Dental Materials. 1st ed. Verlag Berlin Heidelberg; Springer:2009. p.13-40

  1. Ian Freshney Culture of Animal Cells: A Manual of Basic Technique, Fifth Edition,. Haboken; John Wiley & Sons: 2005.p.1-216

Point 2: The numerical values in table 2 are difficult to understand because, for example, ", 002" how to understand it? in one of the columns the values of p were collected, which are explained above as probability. However, in table 2 p takes the values ".691" when it should not exceed 1. All this should be explained and possibly corrected.

Point 4: In table 3, in some places, the values of standard deviations are similar to the obtained values, which should not appear, because it means that the obtained values are marked with too much error ... 

Response 2-4: There are 3 factors in the experiment. 1: AS and ASM, 2: different materials (negative control, T, A, B, P, O, M, S, I, V, positive control). 3: the time factor. The total number of application combinations is 11*2*3=66. Results were obtained using a 5 time dwell time. Measurements over time were not used on the same experimental unit, but a separate set of subjects was used for each time period. Each combination was repeated 3 times. Therefore, there are (2x11x5)x3 =330 observations in the experiment. The trial was set up according to the factorial trial order. The measured property is the absorbant value that defines the toxic effect. The aim is to reveal the effect of melatonin on these absorbant values, the effect of the materials used, the effect of time and the interaction effects of melatonin-material, melatonin-time, material-time and melatonin-material-time. This was described in 2.5. Statistical Analysis. 

Table 2 was revised and expanded. Error andd corrected total description were added.

Figure 1 and 2 were added to be more descriptive.

Point 3: Editorial errors should be corrected throughout the manuscript.

Response 3: The manuscipt was checked and corrected.

Point 5: The Conclusions part is far too laconic and must be expanded to include the concrete achievements of this manuscript.

Response 5: The conlusons part is revised and expanded.

Point 6: The most recent items are missing in the cited literature, for example, at least from the last 3 years. This should be completed.

Response 6: Updated references were added to the manuscript.

Reviewer 2 Report

The article seems to be interesting in the field of dental materials. The Authors assessed the influence of melatonin on the cytotoxicity of some dental materials containing methacrylate. Because melatonin displays antioxidative properties, and several cytotoxic and genotoxic effects of dental methacrylate monomers promote oxidative processes, melatonin may be, due to its antioxidative mechanisms, a protective agent against these effects.

Still, a minor revision should be performed on the manuscript.

Abstract

Line 5 – typo

Line 13 – what is „M”?

Line 14 - typo

Introduction

Line 28 – sentence without sense

Line 37 – „monomers have may cause...” – no sense

Line 48 – repeated words and wrong sentence formulation

M&m

In table 1, it must be specified the way of polymerization for all the studied materials, not only for V, O, I, P, M (e.g. self cure, light cure etc)

Line 89 – what means „24” after CaCl2?

Results

In table 2, commas needs to be replaced with points  

„Cytotoxicity results showed no significant difference between the AS and ASMM  groups (F=0.796; p=0.373). However, there was a significant difference for both tested materials at each incubation period (p<0.001).” – please be more explicit. What is ASMM and what do you mean by „both tested materials”?

Discussion

Line 163 – typo

Line 196 – typo

Some new references should be used. In the manuscript, the newest is from 2015 (and is just one, the others are older)

Author Response

Response to Reviewer 2 Comments

Point 1. Abstract Line 5 – typo

Deleted

Point 2. Line 13 – what is „M”?

Corrrected.

Point 3. Line 14 – typo

Corrrected.

Point 4. Introduction Line 28 – sentence without sense

Corrrected.

Point 5. Line 37 – „monomers have may cause...” – no sense

Corrrected.

Point 6. Line 48 – repeated words and wrong sentence formulation

Revised.

Point 7. M&m

In table 1, it must be specified the way of polymerization for all the studied materials, not only for V, O, I, P, M (e.g. self cure, light cure etc)

The materials brand names were rechecked and only the brand names were written.

Point 8.  Line 89 – what means „24” after CaCl2?

Th referece mismash was deleted.

Point 9. Results In table 2, commas needs to be replaced with points 

They were corrrected.

Point 10. “Cytotoxicity results showed no significant difference between the AS and ASMM  groups (F=0.796; p=0.373). However, there was a significant difference for both tested materials at each incubation period (p<0.001).” – please be more explicit. What is ASMM and what do you mean by „both tested materials”?

ASMM was corrected as ASM. Table 2 was revised and figure 1 and 2 were added.

Point 11. Discussion Line 163 – typo

Deleted.

Point 12. Line 196 – typo

Corrected.

Point 13. Some new references should be used. In the manuscript, the newest is from 2015 (and is just one, the others are older)

Response : Updated references were added to the manuscript.

Reviewer 3 Report

First of all, I would like to thank the Editor for the opportunity to evaluate the following manuscript: Cytotoxicity of acrylic resins, particulate filler composite resin 1 and thermoplastic material in artificial saliva with and without 2 melatonin ”.

The aim of the study was to investigate in vitro the cytotoxic effects of heat- and auto-polymerized acrylic resin, particulate filler composite resin and a thermoplastic material on L-929 fibroblast cell viability at different incubation periods in artificial saliva without and with melatonin.

In my opinion, this study does not provide any new insight into the existing literature. Indeed, a literature review titled  “Perspectives on the use of melatonin to reduce cytotoxic and genotoxic effects of methacrylate-based dental materials” by Janusz Blasiak, Jacek Kasznicki, Jozef Drzewoski, Elzbieta Pawlowska, Joanna Szczepanska and Russel J. Reiter , has been published in  J. Pineal Res. 2011; 51:157–162, summarizing several similar studies. However, in the present work the authors suggest to test the cytotoxic effects of different dental materials (Acrylic resin, Particulate filler resin composite, and Thermoplastic material) on L-929 fibroblast cell viability, providing a little missing piece to the literature.

A major concern arises from the cell model used. Because such investigated monomers may penetrate the pulp due to the incompleteness of polymerization and polymer degradation, why authors decide to use L-929 fibroblast? Why did they not investigate directly the effects of these materials on dental pulp cells?

In my opinion, considering the high requirements of this journal, this manuscript does not have the necessary standards to recommend publication on "Materials".

Author Response

Response to Reviewer 3 Comments

Point 1. In my opinion, this study does not provide any new insight into the existing literature. Indeed, a literature review titled  “Perspectives on the use of melatonin to reduce cytotoxic and genotoxic effects of methacrylate-based dental materials” by Janusz Blasiak, Jacek Kasznicki, Jozef Drzewoski, Elzbieta Pawlowska, Joanna Szczepanska and Russel J. Reiter , has been published in  J. Pineal Res. 2011; 51:157–162, summarizing several similar studies. However, in the present work the authors suggest to test the cytotoxic effects of different dental materials (Acrylic resin, Particulate filler resin composite, and Thermoplastic material) on L-929 fibroblast cell viability, providing a little missing piece to the literature.

Response 1: Blasiak et al., opened a new possible application of meltonin to improve properties of biomaterials used in dentistry. We think this study will add contributions to the dental literature because of novel subject melatonin was applied on different dental resin materials that have different polymerization types and different chemical compositons.

Point 2. A major concern arises from the cell model used. Because such investigated monomers may penetrate the pulp due to the incompleteness of polymerization and polymer degradation, why authors decide to use L-929 fibroblast? Why did they not investigate directly the effects of these materials on dental pulp cells?

Response 2: The L 929 culture we use is sold commercially and is used for viability assessment and live cell detection.

"Two types of cells are used in the cell culture studies. These are primary cells and continuous cell lines. Primary cell cultures are obtained by culturing cells separated from tissues and organs for more than 24 hours. Pulp fibroblasts are examples of the primary cultured cells. However, it is very difficult to isolate and culture primary cultures from humans. Because primary cultures are taken from different individuals, they reflect functional states differently.

Persistent cell lines are transformed primary cells which are capable of proliferating indefinitely and have a more stable phenotype. Continuous cells cannot maintain all their in vivo properties due to the transformation that occurs. Continuous cell lines can be easily propagated. L 929 is also one of the continuous cell lines frequently used in the research studies. Mouse fibroblast cell culture L-929 is used in many studies as a standardized cell culture for the evaluation of viability and determination of viable cell number. It is also possible to repeat cytotoxicity tests because it is standardized. In addition, fibroblast cell cultures are recommended in the standards such as ISO 7405."

Dental Materials 18 (2002) 318-323 Responses of L929 moue fibroblasts, primary and immortalized bovine dental papillaa,derived cell lines to dental resin components. B. Thonemnn, G. Schmalz, K.A. Hiller, H. Schweikl.

ISO7405

Med Oral Patol Oral Cir Bucal 2007;12:E258-66

Schmalz G, Dorthe Arenholt Bindslev. Biocompatibility of Dental Materials. 1st ed. Verlag Berlin Heidelberg; Springer:2009. p.13-40

  1. Ian Freshney Culture of Animal Cells: A Manual of Basic Technique, Fifth Edition,. Haboken; John Wiley & Sons: 2005.p.1-216

Round 2

Reviewer 1 Report

I recommend publishing the manuscript in the current version.

Reviewer 3 Report

Authors  answered  to my requests, therefore, I think this manuscript could be acceptable in ‘Materials'.